# Neutrophil-to-Lymphocyte Ratio and Symptomatic Hemorrhagic Transformation in Ischemic Stroke Patients Undergoing Revascularization

**DOI:** 10.3390/brainsci10110771

**Published:** 2020-10-23

**Authors:** Milena Świtońska, Natalia Piekuś-Słomka, Artur Słomka, Paweł Sokal, Ewa Żekanowska, Simona Lattanzi

**Affiliations:** 1Department of Neurosurgery and Neurology, Nicolaus Copernicus University in Toruń, Ludwik Rydygier Collegium Medicum, 85-168 Bydgoszcz, Poland; pawel.sokal@cm.umk.pl; 2Department of Inorganic and Analytical Chemistry, Nicolaus Copernicus University in Toruń, Ludwik Rydygier Collegium Medicum, 85-089 Bydgoszcz, Poland; natalia.piekus@cm.umk.pl; 3Department of Pathophysiology, Nicolaus Copernicus University in Toruń, Ludwik Rydygier Collegium Medicum, 85-094 Bydgoszcz, Poland; artur.slomka@cm.umk.pl (A.S.); zhemostazy@cm.umk.pl (E.Ż.); 4Neurological Clinic, Department of Experimental and Clinical Medicine, Marche Polytechnic University, 60121 Ancona, Italy; alfierelattanzisimona@gmail.com

**Keywords:** ischemic stroke, hemorrhagic transformation, neutrophil-to-lymphocyte ratio

## Abstract

Objectives: Symptomatic hemorrhagic transformation (sHT) is a life-threatening complication of acute ischemic stroke (AIS). The early identification of the patients at increased risk of sHT can have clinically relevant implications. The aim of this study was to explore the validity and accuracy of the neutrophil-to-lymphocyte ratio (NLR) in predicting sHT in patients with AIS undergoing revascularization. Methods: Consecutive patients hospitalized for AIS who underwent intravenous thrombolysis, mechanical thrombectomy or both were identified. The NLR values were estimated at admission. The study endpoint was the occurrence of sHT within 24 h from stroke treatment. Results: Fifty-one patients with AIS were included, with a median age of 67 (interquartile range, 55–78) years. sHT occurred in 10 (19.6%) patients. Patients who developed sHT had higher NLR at admission. NLR was an independent predictor of sHT and showed good discriminatory power (area under the curve 0.81). In a multivariable analysis, NLR and systolic blood pressure were independently associated with sHT. Conclusions: NLR at admission can accurately predict sHT in patients with AIS undergoing revascularization.

## 1. Introduction

Stroke is one of the main causes of death and long-term disability in adults [1] and the prediction of outcome represents a major clinical goal and research field. Intravenous thrombolysis (IVT) and mechanical thrombectomy (MT) are currently the standard treatments for acute ischemic stroke (AIS) [2,3]. They can improve the functional recovery of patients, but they are also associated with an increased risk of bleeding [4,5,6,7].

Hemorrhagic transformation (HT) is a complication of AIS, and it can be part of the natural course and evolution of ischemic lesions and may be also induced or favored by reperfusion therapy. Hemorrhagic transformation occurs in about 10% to 40% of the patients with AIS and can range from minor petechial bleeding (hemorrhagic infarct) to major, mass-producing hemorrhage (parenchymal hematoma) [8]. Recent IVT trials have reported a rate of any HT up to 46% [9]. Symptomatic HT (sHT) occurs less frequently, and an incidence of 4.4% was reported in the meta-analysis of the Highly Effective Reperfusion evaluated in Multiple Endovascular Stroke Trials (HERMES) collaboration [7]. Symptomatic hemorrhagic transformation is associated with worsening of neurological status, results in prolonged hospital stay and increases the risk of mortality.

The early identification of the patients at increased risk of sHT after cerebral infarction can have practical implications. Factors already associated with sHT include advanced age, massive cerebral infarct involving the middle cerebral artery (MCA) territory, high National Institutes of Health Stroke Scale (NIHSS) score, high blood pressure (BP), hyperglycemia, and treatment with antiplatelets or oral anticoagulants [10,11,12,13,14]. Although preliminary reports have described risk factors for HT after IVT, data remain not conclusive [15]. Furthermore, there is a gap in the understanding of risk factors for sHT when MT is used, alone or in combination with IVT.

Ischemic stroke elicits a robust activation of the immune system. Within 30 min of focal cerebral ischemia, circulating leukocytes adhere to vascular endothelial cells [16]. Leukocyte adhesion and migration across the vasculature activate a number of signaling cascades (e.g., protein kinase C, focal adhesion kinase) that increase brain–blood barrier (BBB) permeability. In humans, infiltration of matrix metalloproteinase-9 (MMP-9)-positive neutrophils is associated with BBB breakdown, basal lamina type IV collagen degradation, and HT [17].

The neutrophil-to-lymphocyte ratio (NLR) is seen as a systemic marker of subclinical inflammation, and an increased ratio is of prognostic value in several disorders. The association between the NLR and HT in AIS has not been fully explored yet. Previous studies have shown that NLR is a predictor of functional outcome in patients with AIS and intracerebral hemorrhage [18,19,20], but their clinical usefulness in sHT after IVT and MT is unknown.

The aim of this study was to explore the relationships between the total and differential leukocyte counts and the NLR at admission with the occurrence of sHT in patients with AIS who underwent treatment with IVT and/or MT.

## 2. Methods

### 2.1. Study Population

We retrospectively identified consecutive AIS patients who underwent treatment with IVT and/or MT (January 2017 to December 2018) and were hospitalized within 24 h from symptom onset at the Interventional Stroke Treatment Center at the University Hospital No. 2 in Bydgoszcz, Poland. Exclusion criteria were: (1) hemorrhagic stroke, (2) transient ischemic attack, (3) recent myocardial infarction (30 days prior to the study), (4) clinical features of infection, (5) liver or kidney failure, (6) history of cancer, (7) oral steroid therapy, (8) pregnancy, and (9) inability to give informed consent. Patients with subarachnoid hemorrhage, arterial dissection, or procedural intracerebral hemorrhage (i.e., intracerebral hemorrhage associated with the thrombectomy procedure itself, secondary bleeding in the perforation site) were also excluded.

Diagnosis of ischemic stroke was made by the same experienced neurologist (M.S.) and confirmed with computed tomography (CT) and magnetic resonance imaging (MRI). Stroke etiology was determined in accordance with the criteria of the Trial of Org 10,172 in Acute Stroke Treatment (TOAST) and the anatomical extent was categorized according to the classification of the Oxfordshire Community Stroke Project (OCSP) [21,22]. Data about demographics, vascular risk factors, medical history, baseline stroke severity according to the NIHSS score, and admission BP values were collected. The ischemic lesion extension was estimated according to the Alberta Stroke Program Early CT Score (ASPECTS) on head CT performed in an emergency, prior to causal treatment.

The venous ethylenediaminetetraacetic acid (EDTA) blood samples were collected at admission. Peripheral-venous blood was drawn for the assessment of complete blood count (CBC) and included leukocyte, neutrophil, and lymphocyte counts. A complete blood count (CBC) was performed using an automated blood counter (XT-4000i, Sysmex Corporation, Kobe, Japan). The NLR was calculated as the ratio of the absolute neutrophil count to the absolute lymphocyte count.

Patients received IVT and/or MT in accordance with the recommendations of the American Heart Association/American Stroke Association [6]. IVT was performed up to 4.5 h and MT, either alone or in combination with IVT, was performed up to 6 h after the onset of stroke symptoms. A thrombectomy was performed by the same interventional radiologist in patients with a confirmed obstruction in the middle cerebral artery (M1/M2) or basilar artery. The endovascular treatment was performed using a stent retriever. The degree of recanalization was assessed using the Thrombolysis in Cerebral Infarction (TICI) scale [23]. CT and MRI imaging was performed routinely for any patient 24 h after treatment as well as individually in case of neurological deterioration. sHT was defined as parenchymal hematoma (PH) >30% of the infarcted area with a significant space-occupying effect or clot remote from the infarcted area (PH2 according to the criteria of the European Cooperative Acute Stroke Study (ECASS)) [24,25,26] associated with a worsening of the NIHSS score by four or more points within 24 h from stroke onset [27].

### 2.2. Statistical Analysis

The normal distribution of each continuous variable was assessed with the Shapiro–Wilk test. Values were reported as medians (interquartile ranges) and comparisons were made with the Mann–Whitney U test. Categorical variables were reported as frequencies and ratios and compared with Pearson’s χ^2^ tests.

The association of the variables with *p* < 0.05 from a comparison of baseline characteristics and selected variables (age, sex) with the occurrence of sHT was determined using a logistic regression analysis. According to Receiver Operating Characteristics Curve (ROC) analysis, the optimal cut-off value was determined as the point maximizing the Youden function, which is the difference between the true positive rate and false positive rate over all possible cut-off point values. Results were considered significant for *p* values < 0.05 (2 sided). The STATISTICA software was used (version 13.3, TIBCO Software Inc., Palo Alto, CA, USA).

### 2.3. Standard Protocol Approvals, Registrations, and Patient Consents

The study protocol was verified and approved by the Bioethical Commission (KB 694/2016) at the Collegium Medicum UMK (Bydgoszcz, Poland).

### 2.4. Data Availability

Anonymized data will be shared by request from any qualified investigator.

## 3. Results

A total of 281 patients diagnosed with AIS and undergoing causal treatment were admitted. After the exclusion of 230 patients, 51 patients were selected. The reasons of exclusion were (1) evidence of active infection before admission or any systemic infection that occurred during the first 48 h after casual treatment (194 patients); (2) cancer, chronic inflammation, autoimmune disease, or steroid therapy (10 patients); (3) unavailability to complete blood cell count or medical records (26 patients, who were discharged on the same day of admission). Recombinant tissue plasminogen activator (rt-PA) was administered intravenously to 45 patients, 17 of whom also received MT. Six patients underwent MT alone as they exceeded the time window (>4.5 h) for the IVT.

Demographics and clinical characteristics of the study population are summarized in Table 1. The study population consisted of 22 males and 29 females, with a median age of 67 (IQR, 55–78) years. A total of 10 out of 51 (19.6%) patients experienced sHT. As shown in Table 1, patients who presented sHT had higher systolic BP (SBP) and diastolic BP (DBP), higher serum creatinine (sCr), higher numbers of neutrophils (NEUTs) and lower numbers of lymphocytes (LYMPHs) and platelets (PLTs). The NLR was significantly higher (*p* = 0.002) in patients with sHT than those without sHT. There were no significant differences between the subgroups in terms of age, sex, body mass index, use of antiplatelets and oral anticoagulants before stroke, left ventricular ejection fraction function, liver function, low-density lipoprotein, glucose level, white blood cell, red blood cell, monocyte, hemoglobin, hematocrit and blood coagulation parameters as the international normalized ratio and activated partial thromboplastin time.

The stroke characteristics of included patients are summarized in Table 2. Cardioembolism and large artery atherosclerosis were the most common etiologies within the group of patients with sHT, whereas stroke of unknown etiology was prevalent in the non-hemorrhagic group. Patients with sHT showed a more severe neurological deficit at admission as assessed by the NIHSS score (*p* = 0.001). No statistically significant differences were found across the groups in size and location of cerebral infarct, ASPECT scale score at admission, and rate of vascular recanalization. The rate of sHT was significantly higher in patients undergoing IVT combined with MT or MT alone than patients treated with IVT alone. (Table 2).

For the multiple logistic regression analysis, sHT was independently associated with NLR (OR = 1.32, 95% CI 1.07–1.63, *p* = 0.009) and SBP (OR = 1.06, 95% CI 1.01–1.11, *p* = 0.013); no associations were found with age, sex, NIHSS, NEUTs, LYMPHs, DBP, sCR and PLTs.

ROC curves were generated for the variables that were significantly different between patients with and without sHT for a univariate comparison. The details referring to the area under the curve (AUC), the sensitivity, and the specificity for each variable are shown in Table 3.

## 4. Discussion

This study has shown that NLR can be a good predictor of sHT in AIS patients being higher NLR at admission associated with a greater risk of hemorrhagic transformation and neurological worsening within the first 24 h from revascularization with IVT and/or MT.

The search for prognostic factors of ischemic stroke, including serious complications such as hemorrhage, remains a clinical issue. The main goal of this study was to identify a simple indicator that could reliably predict early hemorrhagic transformation after revascularization.

Earlier studies suggested that hemorrhagic transformation itself can have detrimental effects on the clinical course of stroke, and the influence of HT on outcome depends largely on its type. So far, it has been found that parenchymal hematoma type 2 (PH2) can change the clinical course of stroke and represents a significant predictor of neurological deterioration and higher mortality [28,29,30,31]. NLR is a laboratory biomarker that has already been described in many studies. As a marker of stroke-related, acute inflammatory response, it has been shown to be a good predictor of outcomes. In most studies, high NLR values have been associated with a worse functional status of patients after AIS [32,33,34]. However, it should be noted that these studies mainly focused on patients undergoing IVT alone. To the best of our knowledge, this study is one of the first concerning patients treated with IVT combined with MT or MT alone. After ischemic stroke has occurred, tissue hypoxia promotes an inflammatory reaction within brain parenchyma [35]. Circulating neutrophils are recruited to the site of brain damage shortly after ischemia. Neutrophils are among the first cells to penetrate hypoxic tissue, and this happens within the first hours after reperfusion. Neutrophils can cause damage to the BBB and contribute to the injury of surrounding tissues [8,36,37,38]. Recently, Maestrini et al. reported that higher neutrophil counts and NLR before thrombolysis in cerebral ischemia were independently associated with sICH and worse 3-month outcome [32]. Conversely, NLR values before IVT did not significantly predict IV effectiveness and functional status after stroke in the study by Pektezel et al. [39]. Additionally, high NLR values were found to be a good predictor of early neurological deterioration and poor 3-month functional status in patients who had hemorrhagic stroke [40,41,42].

We found significantly higher values of NLR at admission in patients with sHT, where hemorrhage occurred within 24 h from treatment. In this regard, there is evidence suggesting that bleeding occurring within 24 h, and up to 36 h, should be interpreted as a complication of treatment, whereas a hemorrhage that develops after a longer time interval should not be regarded as a direct result of revascularization procedures [43,44]. Remarkably, in building decision models, it is important to determine the optimal cut-off point value. In our research, the best threshold of the NLR at admission for sHT occurrence was 9.68, which is consistent with previous observations [45,46,47,48].

This study does have different limitations. First of all, the sample size was small. Second, the retrospective design and single-center nature of the study could have led to selection bias and limit the generalizability of results. Third, the exact time intervals between the onset of stroke symptoms and the initiation of treatment or blood draw were not available. Further research is justified and required to validate these findings in independent cohorts and explore the dynamic changes of NLR values over time. The main strengths of the study included the use of widely accessible laboratory variables and the cost effectiveness of the NLR.

## 5. Conclusions

In summary, the NLR can be a simple, readily available and inexpensive tool for the early identification of patients at an increased risk of early symptomatic hemorrhage after recanalization in AIS. Further understanding of the key players of the inflammatory response may help to identify therapeutic strategies.

## Figures and Tables

**Table 1 brainsci-10-00771-t001:** Baseline characteristics of patients according to Symptomatic hemorrhagic transformation (sHT).

Parameter (Unit)		Total (*n* = 51)	no-sHT (*n* = 41)	sHT (*n* = 10)	*p* Value
Age (years)		67 (55–78)	63 (54–76)	74 (67–79)	0.138
SEX-male N/total (ratio)		22/51 (0.43)	17/41 (0.41)	5/10 (0.50)	0.631
BMI [kg/m^2^]		27 (24–32)	26 (24–32)	28 (25–31)	0.887
Medical history, N/total (ratio)	CAD	13/51 (0.25)	8/41 (0.20)	5/10 (0.50)	0.055
	Previous AIS	6/51 (0.12)	6/41 (0.15)	0/10	0.198
	Hypertension	38/51 (0.75)	29/41 (0.71)	9/10 (0.90)	0.215
	Diabetes mellitus	16/51 (0.31)	12/41 (0.29)	4/10 (0.40)	0.509
	Dyslipidemia	28/51 (0.55)	24/41 (0.59)	4/10 (0.40)	0.291
	AF	15/51 (0.29)	10/41 (0.24)	5/10 (0.50)	0.118
Anticoagulant therapy N/total (ratio)	VKA	5/51 (0.10)	3/41 (0.07)	2/10 (0.20)	0.231
	Antiplatelets	14/51 (0.27)	12/41 (0.29)	2/10 (0.20)	0.567
Current smokers, N/total (ratio)		13/51 (0.25)	11/41 (0.27)	2/10 (0.20)	0.664
SBP (mmHg)		130 (130–150)	130 (130–150)	158 (140–180)	0.009
DBP (mmHg)		80 (80–90)	80 (80–80)	90 (80- 100)	0.027
LVEF (%)		60 (50–65)	60 (50–65)	60 (55–60)	0.374
AST (U/L)		19 (16–25)	19 (16–24)	20 (17–29)	0.313
ALT (U/L)		17 (13–22)	17 (13–25)	16 (12–19)	0.265
sCr (mg/dL)		0.85 (0.71–1.08)	0.81 (0.69–1.03)	1.07 (0.80–1.15)	0.024
LDL (mg/dL)		89 (73–132)	94 (73–138)	78 (74–119)	0.553
Glucose (mg/dL)		125 (107–151)	122 (109–150)	134 (107–151)	0.537
CBC	WBCs (10^3^/μL)	10.88 (8.70–13.30)	10.75 (8.67–13.05)	12.06 (10.04–17.94)	0.132
	NEUTs (10^3^/μL)	8.65 (6.51–12.49)	8.00 (5.95–9.69)	13.18 (8.52–15.58)	0.034
	LYMPHs (10^3^/μL)	1.64 (1.04–2.13)	1.76 (1.13–2.32)	1.13 (0.75–1.63)	0.012
	MONOs (10^3^/μL)	0.94 (0.71–1.15)	0.94 (0.71–1.13)	1.00 (0.66–1.33)	0.469
	RBCs (10^6^/μL)	4.31 (3.99–4.69)	4.31 (3.99–4.61)	4.33 (4.07–5.06)	0.545
	HGB (g/dL)	13.3 (12.1–14.2)	13.3 (11.9–14.2)	13.1 (12.1–14.3)	0.367
	HCT (%)	38.2 (36.0–42.5)	38.2 (35.9–42.3)	38.8 (36.1–42.7)	0.522
	PLTs (10^3^/μL)	233 (185–263)	241 (206–268)	195 (164–228)	0.040
NLR		5.46 (3.41–9.15)	4.72 (3.00–8.30)	10.39 (7.00–14.98)	0.002
INR		1.1 (1.0–1.2)	1.1 (1.0–1.2)	1.1 (1.1–1.6)	0.155
aPTT		26.4 (25.0–29.8)	26.2 (25.0–29.8)	26.9 (25.1–28.5)	0.831

Data are expressed as median (interquartile range, IQR) or frequencies and ratios. The *p* values were determined using the Pearson’s χ^2^ tests (categorical variables) or the Mann–Whitney U test (continuous variables). Abbreviations: n: number, BMI: body mass index, CAD: coronary artery disease, AIS: acute ischemic stroke, and AF: atrial fibrillation, VKA: vitamin K antagonists. SBP: systolic blood pressure, DBP: diastolic blood pressure, LVEF: left ventricular ejection fraction, AST: aspartate aminotransferase, ALT: alanine aminotransferase, sCr: serum creatinine, LDL: low-density lipoprotein, CBC: complete blood count, WBCs: white blood cells, NEUTs: neutrophils, LYMPHs: lymphocytes, MONOs: monocytes, RBCs: red blood cells, PLTs: platelets, NLR: neutrophil-to-lymphocyte ratio, INR: international normalized ratio, aPTT: activated partial thromboplastin time.

**Table 2 brainsci-10-00771-t002:** Stroke characteristics according to Symptomatic hemorrhagic transformation (sHT).

Parameter (Unit)		Total (*n* = 51)	no-sHT (*n* = 41)	sHT (*n* = 10)	*p* Value
TOAST N/total (ratio)	LAA	11/51 (0.22)	7/41 (0.17)	4/10 (0.40)	0.032
	SVO	5/51 (0.10)	5/41 (0.12)	0/10	
	CE	16/51 (0.31)	10/41 (0.24)	6/10 (0.60)	
	SOE	3/51 (0.06)	3/41 (0.08)	0/10	
	SUE	16/51 (0.31)	16/41 (0.39)	0/10	
OSCP N/total (ratio)	LACI	13/51 (0.25)	13/41 (0.32)	0/10	0.052
	PACI	21/51 (0.41)	17/41 (0.41)	4/10 (0.40)	
	TACI	8/51 (0.16)	4/41 (0.10)	4/10 (0.40)	
	POCI	9/51 (0.18)	7/41 (0.17)	2/10 (0.20)	
Type of treatment N/total (ratio)	Thrombolysis	28/51 (0.55)	28/41 (0.68)	0/10	0.0001
	Thrombolysis and Thrombectomy	17/51 (0.33)	11/41 (0.27)	6/10 (0.60)	
	Thrombectomy	6/51 (0.12)	2/41 (0.05)	4/10 (0.40)	
Baseline NIHSS Initial ASPECTS		11 (6–16)	8 (4–14)	17 (14–20)	0.001
		10 (9–10)	10 (9–10)	9 (9–9)	0.173
TICI	TICI0	2/23 (0.09)	1/13 (0.08)	1/10 (0.10)	0.249
	TICI2b	9/23 (0.39)	7/13 (0.54)	2/10 (0.20)	
	TICI3	12/23 (0.52)	5/13 (0.38)	7/10 (0.70)	

Data are expressed as median (interquartile range, IQR) or frequencies and ratios. *p* values were determined using the Pearson’s χ^2^ tests (categorical variables) or the Mann–Whitney U test (continuous variables). Abbreviations are TOAST: Trial of ORG 10,172 in Acute Stroke Treatment, LAA: large-artery atherosclerosis, SVO: small-vessel occlusion, CE: cardioembolism, SOE: stroke of other determined etiology, SUE: stroke of undetermined etiology, OSCP: Oxfordshire Community Stroke Project, LACI: lacunar circulation infarcts, PACI: partial anterior circulation infarcts, TACI: total anterior circulation infarcts, POCI: posterior circulation infarcts, NIHSS: National Institutes of Health Stroke Scale, mRS: Modified Rankin Scale, ASPECTS: Alberta Stroke Program Early CT Score, TICI: thrombolysis in cerebral infarction.

**Table 3 brainsci-10-00771-t003:** Details of the receiver operating characteristics (ROCs) analysis.

Parameter (Unit)	AUROCC	95% CI	Cut-Off	Sensitivity	Specificity
SBP (mmHg)	0.771	0.587–0.954	140	0.800	0.659
DBP (mmHg)	0.729	0.548–0.910	90	0.600	0.805
Baseline NIHSS	0.837	0.717–0.956	12	0.900	0.683
sCr (mg/dL)	0.733	0.586–0.880	0.78	1.000	0.488
PLTs (10^3^/μL)	0.712	0.544–0.880	234	0.900	0.561
NEUTs (10^3^/μL)	0.720	0.536–0.903	12.49	0.600	0.829
LYMPHs (10^3^/μL)	0.761	0.602–0.920	1.98	1.000	0.390
NLR	0.815	0.667–0.962	9.682	0.600	0.927

Abbreviations: AUROCC: Area Under the Receiver Operating Characteristics Curve, SBP: systolic blood pressure, DBP: diastolic blood pressure, NIHSS: National Institutes of Health Stroke Scale, sCr: serum creatinine, NEUTs: neutrophils, LYMPHs: lymphocytes, PLTs: platelets, NLR: neutrophil-to-lymphocyte ratio.

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
