# Peer review of "Neutrophil-to-Lymphocyte Ratio and Symptomatic Hemorrhagic Transformation in Ischemic Stroke Patients Undergoing Revascularization"

_brainsci, 2020, doi:10.3390/brainsci10110771_

Round 1

Reviewer 1 Report

All round well written manuscript. Authors present data indicating NLR is a predictor of haemorrhagic transformation post-endovascular therapies. 

Below are recommended comments to be addressed. 

Minor

Line 47: would be beneficial to define the threshold that constitutes a "massive cerebral infarct". 

Major

It was not clear if time between stroke symptom onset to endovascular therapy or even blood draw was declared or controlled for in the multivariate analysis. From the persepective of this study, the time from symptom onset to endovascular therapy would also be an important indicator of potential haemorrhagic transformation - the longer the MCAO (M1/M2), the greater propensity for BBB degradation and haemorrhagic transformation post-revascularisation.

In addition the time from stroke symptom onset to blood draw should be controlled for as length of vessel occlusion is likely to impact clinical findings in blood panels. 

Author Response

REPLY TO REVIEWER #1 COMMENTS

We appreciate your interest in our manuscript and the most valuable suggestions to improve its quality. All the changes are highlighted in yellow in the revised text.

Below, we enclose a point-by-point response to your comments.

  1. Line 47: would be beneficial to define the threshold that constitutes a “massive cerebral infarct”.

Reply: Thank you very much for your comment. The definition of a “massive cerebral infarct” has been added.

  1. It was not clear if time between stroke symptom onset to endovascular therapy or even blood draw was declared or controlled for in the multivariate analysis. From the persepective of this study, the time from symptom onset to endovascular therapy would also be an important indicator of potential haemorrhagic transformation - the longer the MCAO (M1/M2), the greater propensity for BBB degradation and haemorrhagic transformation post-revascularisation.

In addition the time from stroke symptom onset to blood draw should be controlled for as length of vessel occlusion is likely to impact clinical findings in blood panels.

Reply: Thank you for bringing this to our attention. Unfortunately, the exact time between the onset of stroke symptoms and the initiation of treatment has not been precisely defined. During the period when the patients were being treated in our clinic, this time was not precisely defined, however, thrombolysis was performed up to 4.5 hours and thrombectomy, either alone or in combination with thrombolysis, up to 6 hours after the onset of symptoms. This limitation is now presented in the current version of the manuscript. It is a very valuable comment for us, and this data will be collected when planning further research.

Again, thank you for your comments and suggestion.

Reviewer 2 Report

The manuscript entitled “Neutrophil-to-lymphocyte ratio and symptomatic 2 hemorrhagic transformation in ischemic stroke 3 patients undergoing revascularization” by Milena and colleagues studies the prediction of sHT after thrombolysis or thrombectomy with the NLR. The manuscript is well-written. I have some suggestions and concerns about the study.

  1. Hyperglycemia is also a risk factor for developing HT, please add this to the introduction part.
  2. on page 2, line 53, “Within 30 minutes of focal cerebral ischemia, circulating leukocytes adhere to vascular endothelial cells.”, please provide citation to this sentence.
  3. In the discussion part, the author mentioned that “and only this type of hemorrhage can change the clinical course of stroke [25- 28].” I think this may not be true. Some studies suggest that even minor HT can have negative effects on stroke outcomes. Please double check.
  4. In the keyword, neutrophil-to-lymphocyte ratio and NLR are the same, I would suggest put them together.
  5. The authors indicate that “The rate of sHT was significantly higher in patients undergoing IVT combined with MT or MT alone than patients treated with IVT alone.” What is the time point that patients received the MT or IVT? Is it possible that the MT group usually was treated in a wider time window than the IVT group? Please show the data.
  6. Did the authors observe any changes in NLR after IVT and/or MT treatment?

Author Response

REPLY TO REVIEWER #2 COMMENTS

We would like to thank you for your insightful and valuable comments which have greatly helped us to improve the quality of the manuscript. All the changes are highlighted in yellow in the revised text.

Below, we enclose a point-by-point response to your comments.

  1. Hyperglycemia is also a risk factor for developing HT, please add this to the introduction part.

Reply: Thank you for your comment. Hyperglycemia as a risk for HT has bee added to the introduction.

  1. On page 2, line 53, “Within 30 minutes of focal cerebral ischemia, circulating leukocytes adhere to vascular endothelial cells.”, please provide citation to this sentence.

Reply: We are very grateful to the Referee for pointing this out. The citation has been added in the revised version of the manuscript.

  1. In the discussion part, the author mentioned that “and only this type of hemorrhage can change the clinical course of stroke [25- 28].” I think this may not be true. Some studies suggest that even minor HT can have negative effects on stroke outcomes. Please double check.

Reply: Thank you for your constructive observation. The sentence has been corrected accordingly.

  1. In the keyword, neutrophil-to-lymphocyte ratio and NLR are the same, I would suggest put them together.

Reply: Thank you for your comment. The keyword has been changed accordingly.

  1. 5. The authors indicate that “The rate of sHT was significantly higher in patients undergoing IVT combined with MT or MT alone than patients treated with IVT alone.” What is the time point that patients received the MT or IVT? Is it possible that the MT group usually was treated in a wider time window than the IVT group? Please show the data.

Reply: We thank the Reviewer for drawing this to our attention. The sentence “IVT was performed up to 4.5 hours and MT, either alone or in combination with IVT, up to 6 hours after the onset of ischemic stroke symptoms.” has been added to the revised text.

  1. Did the authors observe any changes in NLR after IVT and/or MT treatment?

Reply: Thank you for the question. In this study, we only evaluated the NLR at admission to hospital. We have not examined the dynamics of changes in this parameter, and we acknowledged this issue among the limits of the study. We thank the reviewer for this constructive comment, which we also consider as a valuable hint for subsequent studies.

Round 2

Reviewer 1 Report

Authors have addressed my original suggestions and amended the limitations section of the manuscript. 

Reviewer 2 Report

Thanks for the responses